# An Improved Game Theory-Based Cooperative Localization Algorithm for Eliminating the Conflicting Information of Multi-Sensors

**DOI:** 10.3390/s20195579

**Published:** 2020-09-29

**Authors:** Chao Tang, Lihua Dou

**Affiliations:** 1School of Automation, Beijing Institute of Technology, Beijing 100081, China; 3120170434@bit.edu.cn; 2Beijing Institute of Technology Chongqing Innovation Center, Chongqing 401135, China

**Keywords:** Kalman filter, information conflict, SVD, game theory, co-localization

## Abstract

In this article, an improved game theory-based co-localization algorithm is proposed to precisely and cooperatively locate the multi-robot system in the wireless sensor network and efficiently eliminate the information conflict caused by multi-sensor. Specifically, the extended Kalman filter in the original algorithm is replaced by the unscented Kalman filter in the optimized algorithm, which contributes to lower linearization errors and higher localization precision. Then, the computational complexity is analyzed, and the derivative method is introduced to reduce the extra computation burden brought by the unscented Kalman filter. Subsequently, the stability issue resulting from the derivative method is addressed by introducing the singular value decomposition (SVD). In this context, the optimized algorithm is capable of precisely locating the multi-robot system, while maintaining the stability and not increasing the computational burden. Moreover, as demonstrated by the simulation results, the optimized algorithm has greater localization precision than the original algorithm, while they have similar computational burdens.

## 1. Introduction

The multi-robot system has been proven to have better stability, stronger robustness, and lower redundancy than the single-robot system. Consequently, it has been widely applied in large-scale searches, post-disaster searches, and alert patrols, where all robots are expected to be fully aware of their locations in order to efficiently perform cooperative tasks. In this context, precise localization is critical and requires comprehensive investigation.

There are generally three steps in the traditional autonomous localization methods when it comes to sensor information processing. The first step is to collect the information obtained by both internal and external sensors [1,2]; the second step is to process the collected information and ensure that it conforms to the filtering algorithm’s format; the last step is to calculate the localization result based on the information processed by the filtering algorithm [3]. In the case of a multi-robot system, the localization methods are usually extended from those designed for the single-robot system. For example, Roumeliotis et al. [4] distributed the interaction term to the relevant robots in the extended Kalman filtering (EKF) algorithm for multi-robot co-localization. Zhang et al. [5] proposed a cooperative localization approach with communication delays, using the EKF technique based on state estimation, and they suggested reducing the state estimation error of delay filtering based on the error compensation. Wang et al. [6] implemented the co-localization for wireless sensor networks (WSNs) by using a hard decision-based method with outlier constraints in the cooperative localization phase, which helped successfully solve the issue of range-free estimation and effectively avoided large errors in node localization. In order to provide positioning services in larger area, P.N. Pathirana et al. [7] adopted the method of multiple robots touring around the WSNs and used the robust EKF to eliminate measurement noise. However, this method has a large delay and poor performance in real-time. A. Baggio et al. [8] improved the Monte Carlo localization algorithm in WSNs. They optimized the performance of unknow nodes calling position information. H. Chen et al. [9] studied the localization issues in WSNs with physical obstacles. They adopted the convex optimization method to reduce the moving distance of mobile elements in the networks, which extended the life of the mobile elements, while ensuring high-precision localization. Guo et al. [10] suggested supporting the co-localization of unmanned aerial vehicles (UAVs) in a non-GPS environment by adopting an Ultra-Wide Band (UWB)-based EKF localization algorithm. Pirník et al. [11] investigated the performances of inertial devices in controlling wheeled mobile platforms and developed an algorithm to integrate inertial sensor-recorded data into the system control. In this context, they experimentally proved the necessity of the inertial devices to interact with other absolute sensors. Liu et al. [12] combined particle filtering (PF) with the particle swarm optimization algorithm to deal with the co-localization of heterogeneous robot systems.

However, all the dynamic models in the abovementioned studies are of simple trajectories of motion, which means that they are difficult to apply in actual situations. Accordingly, efforts have been made to fill the gap by some European scholars [13]. For instance, Božek et al. [14] used the Newton–Euler method to formulate a dynamic model of a six-axis hexacopter equipped with a robotic arm, and a dynamic model for controlling a four-wheeled highly-dynamic robot was proposed by Kilin et al. [15]. In addition, C. Yu et al. did some studies on identification of state-space in the case of unknown dynamic model [16,17] or unknown input [18]. These studies made great contributions to filtering for complex dynamic models.

Although the above-mentioned methods manage to effectively solve the co-localization problem of the multi-robot system, they are frequently vulnerable to conflicting relative observation information that would lead to divergence in the Kalman filtering (KF) and increase the computational burden. When extending the localization algorithms from single-robot systems to multi-robot counterparts, it is of critical importance to fuse the relative observation information between multiple robots. Otherwise, the possibly present conflicts among different robots in terms of the relative observation information could lead to inconsistent co-localization and even filter divergence. Taking this into consideration, Hua et al. [19] integrated the complete information static game theory into the EKF algorithm for both one-way and two-way observations in the multi-robot systems, which suppressed the observation conflicts significantly.

However, the EKF algorithm adopted by Hua et al. [19] is of low precision, and its improvement is the main focus of this study. Meanwhile, efforts are made to not incur other side effects (e.g., computational burden and instability). In this respect, the work of this study is summarized as follows:The unscented Kalman filter (UKF) algorithm is demonstrated to outweigh EKF in terms of precision [19]. Therefore, UKF is adopted for co-localization in this research.The extra computational burden brought by UKF is reduced by introducing the derivative method [20].Singular value decomposition (SVD) is utilized to enhance the stability of the derivative UKF.The performance of the proposed algorithm is demonstrated by a series of simulation examples.

Finally, a high-precision co-localization algorithm is obtained, which is more effective than the original algorithm in eliminating information conflicts caused by multiple sensors, with similar stability and computational burden. This is the major contribution of this article.

The rest of the paper is organized as follows. In Section 2, a game theory-based algorithm is introduced for co-localization, accompanied by the analysis of its filtering component. Moreover, attention is paid to investigating the KF series algorithms and the UKF-based derivative method as well as comparing UKF with EKF. In Section 3, the dynamic model of the system is established, and an estimation fusion algorithm is proposed based on the derivative UKF in the game theory-based co-localization algorithm framework. Specifically, KF is used to simplify the calculation of the measurement update without reducing the estimation precision. In addition, SVD is applied to the derivative UKF (DUKF) algorithm to improve the stability of the algorithm and to avoid ill-posed problems during Cholesky decomposition. The game theory-based co-localization algorithm is optimized with the proposed filter algorithm and provides the specific steps of the optimized algorithm in Section 4. In Section 5, the optimized algorithm is verified by numerical simulations of the motion system. Eventually, conclusions are drawn in Section 6.

The relationship between these filtering algorithms is shown in Figure 1.

## 2. Relevant Work and Problem Formulation

### 2.1. The Co-Localization Based on the Game Theory and the Cooperative Observation

Hua et al. [19] integrated the complete information static game theory into the EKF algorithm for both one-way and two-way observations in the multi-robot systems, which significantly relieved the computational burden and observation conflicts.

A multi-robot system generally outperforms a single-robot system in the non-GPS environment, when it comes to the relative information observation. For example, when robot *A* appears in the view range of robot *B* or two robots meet each other face to face, they will obtain the observation information concerning the location of the other. The co-localization algorithm based on the static game theory utilizes the relative observation information among multiple robots to achieve more precise localization. During the application of the algorithm, some assumptions have to be imposed on these robots:The location and direction of any robot can be described by a vector in a unified global coordinate system.Each robot is equipped with sensors to perform autonomous localization and observe other robots in the non-GPS environment.All the robots can instantly communicate with each other by sending their own information and observation information.All the robots are isomorphic, which means that they share exactly the same type of observations that can be described with the same mathematical model.

As shown in Figure 2, the vector $\left[\Delta d^{\left(ij\right)},\alpha^{\left(ij\right)}\right]$ represents the pose information when robot *A* observes robot *B*, where $\Delta d^{\left(ij\right)}$ and $\alpha^{\left(ij\right)}$ are the distance and the relative angle between robots *A* and *B*, respectively. Similarly, the vector $\left[\Delta d^{\left(ji\right)},\alpha^{\left(ji\right)}\right]$ represents the pose information when robot B observes robot A. In ideal conditions, the relative distances and the angles observed by the two robots are the same. However, there are inevitable noises in the environment, which result in imprecise observations. In this paper, all noises are assumed to be white noises that follow a normal distribution. Since all robots are isomorphic, $\sigma^{\rho }$ is adopted to represent the distance noise, while $\sigma^{\alpha }$ stands for the angle noise.

The observation by sensors is vulnerable to the effects of noises, which will lead to information conflict among the observations of robots. Accordingly, static game theory is introduced to select optimal information when there are conflicts.

Suppose that $G=\left\{1,2,\ldots ,n\right\}$ is the set of all robots. Each robot corresponds to a separate number. The strategy space of the i-th robot is $s_{i}=\left\{0,1\right\}$, where 1 indicates the independent location and 0 represents the co-location. In the game theory model, the benefit of an individual participant depends on the chosen localization strategy by all participants. When other robots adopt strategies $\left(s_{1},\ldots ,s_{i-1},s_{i+1},\ldots ,s_{n}\right)$ and the robot *i* adopts strategy $s_{i}$, the benefit function is defined as
(1)$$u_{i}\left(s_{1},\ldots ,s_{i},\ldots ,s_{n}\right).\;$$

When *n* robots perform co-localization, the benefit function can be represented by $G=\left\{s_{1},\ldots ,s_{n},u_{1},\ldots ,u_{n}\right\}$ [21].

### 2.2. Problem Formulation

The algorithm by Hua et al. [19] can solve the observation conflicts of co-localization in multi-robot systems. However, its filtering component adopts a conservative technology, which is accompanied by a compromise in precision. As a countermeasure, UKF, instead of EKF in the original algorithm, is employed to improve the precision of the co-localization. Nevertheless, UKF is also not a perfect algorithm, because it has some inherent defects, such as high computational burden and low system stability. To conclude, the following three measures should be taken to improve the performance of the algorithm by Hua et al. [19]:EKF in the original algorithm should be replaced by UKF to improve localization precision.The computational burden of UKF should be relieved.The algorithm stability should be maintained.

Theoretically, the objectives of this study are mainly threefold:to improve the precision of the algorithm by Hua et al. [19] by introducing UKF;to relieve the computational burden of UKF without compromise in precision by introducing the derivative method;to solve the non-positive definite variance matrix caused by the derivative method by replacing the Cholesky decomposition with SVD.

## 3. Improvement for the UKF

### 3.1. Relevant Work on the Kalman Filtering

The KF series algorithms have attracted extensive attention from the engineering community since they were adopted by NASA to solve the navigation problem in the Apollo moon-landing program. However, the traditional KF algorithm is exclusively suitable for linear systems; consequently, it fails to solve problems in nonlinear systems that are ubiquitous in reality (e.g., state estimation and system identification). In this context, nonlinear filtering algorithms are urgently needed.

Recently, a large number of nonlinear filtering algorithms have been proposed, such as the Extended Kalman Filtering (EKF), the Gaussian Sum Filtering, the Gauss–Hermite Filtering, the Particle Filtering (PF), and the Unscented Kalman Filtering (UKF). Among them, EKF is the most commonly used in research on localization and navigation. Its main idea is to first approximately linearize the nonlinear functions and then apply the traditional KF algorithm to filter the linearized system. Although the EKF algorithm is very mature and widely used, it eliminates high-order terms when linearizing a system by the Taylor expansion, thereby leading to truncation errors and ultimately affecting the filtering precision. Moreover, the complexity in calculating the associated Jacobian matrix hinders the wide application of the EKF algorithm in engineering practice [20].

In order to avoid the problem of insufficient accuracy caused by the process of linear approximation, Julier and Uhlmann [22] proposed the UKF algorithm to approximate the posterior probability density function of the nonlinear system based on a set of sigma points by the unscented transformation (UT).

The UKF outperforms the EKF in the following aspects:UKF does not require calculating the Jacobian matrix using the nonlinear state function and the measurement function, which makes it easy to implement in the engineering and simulation.Regardless of the degree of nonlinearity of the system, the posterior mean and variance approximated by UT are equivalent to those of a nonlinear Gaussian system approximated by a third- or higher-order Taylor expansion [20]. In other words, UKF presents a higher precision than EKF in high-order systems.

Based on the above advantages, we believe that the accuracy of localization will be improved by using the UKF algorithm for filtering, which can also reduce information conflicts among multiple robots during the process of co-localization. The flow and step of the game theory-based UKF co-localization algorithm has been given in our previous work [23], so they are not repeated here.

### 3.2. Improvement of the Computational Burden Based on the Derivative Method

Compared with EKF, the high precision of UKF is unfortunately accompanied by a significantly increased computational burden. As a countermeasure, UKF is properly modified in this study by introducing the derivative method.

In the original algorithm, Hua et al. [19] adopted X=[dcosθ,dsinθ] to describe the state of the system. Instead of this, $X={\left[x,y,v_{x},x_{y}\right]}^{T}$ is used for such a description in this study. In this way, the time update can be achieved by the classical KF. Nonlinear filtering is adopted for the observation update based on UT.

The derivative UKF (DUKF) algorithm has the same form as the classical KF algorithm in terms of the time update. This allows DUKF to avoid UT of the time update, which is required in the traditional UKF. As a consequence, the state estimation and its covariance can be calculated concisely, thus reducing the computational burden efficiently. In the observation update phase, the DUKF algorithm relies on the traditional UKF to update the predicted observations, thereby inheriting the excellent performance of the traditional UKF algorithm for nonlinear systems.

### 3.3. System Model Establishment

The considered nonlinear discrete-time system is represented by
(2)$$X_{k+1}=\varphi X_{k}+w_{k}=\left[\begin{array}{cccc} 1 & t & 0 & 0\\ 0 & 1 & 0 & 0\\ 0 & 0 & 1 & t\\ 0 & 0 & 0 & 1 \end{array}\right]X_{k}+w_{k},\;$$
(3)$$Z_{k}=H_{k}\left(X_{k}\right)+v_{k}=\left[\begin{array}{c} r_{k}^{1}\\ r_{k}^{2}\\ \ldots \\ r_{k}^{i} \end{array}\right]+v_{k},\;$$
where $X_{k}$ and $Z_{k}$ denote the state and measurement vectors at the time *k*, respectively, φ is the state prediction matrix, *t* is the sampling time, $H_{k}\left(.\right)$ is the nonlinear measurement function, and $r_{k}^{i}$ is the distance between the *i*-th UWB sensor and the robot at time *k*. To increase the real-time performance of the system, we adopt the distance *r* instead of the coordinates *x* and *y* as the observation value in the measurement vector [24]. $r_{k}^{i}$ is defined as
(4)$$r_{k}^{i}=\sqrt{{\left(x_{i}-x_{k}\right)}^{2}+{\left(y_{i}-y_{k}\right)}^{2}}.\;$$

The $w_{k}$ and $v_{k}$ are the uncorrelated zero-mean Gaussian white noise processes, with covariances as (5)$$\begin{array}{l} E\left[w_{k}w_{k}^{T}\right]=Q_{k}\\ E\left[v_{k}v_{k}^{T}\right]=R_{k}.\; \end{array}$$

It can be seen from the system model that the system has a linear time update matrix and a nonlinear measurement function. Therefore, it meets the conditions required for the application of the DUKF algorithm.

To transplant the algorithm to the physical platform in the future, the values of error variance and system state are determined according to the robot and sensors we purchased. Specifications of simulated sensors and mobile robots are shown in Figure 3. Each base station is equipped with one UWB sensor to measure the distance, and each robot is equipped with three UWB sensors that form an antenna array to obtain the azimuth information during the relative observation.

### 3.4. Comparison between UKF and DUKF in Terms of Computational Complexity

To evaluate the computational burden of the identification algorithm, we count the number of multiplication and addition operations of the algorithm. Each multiplication or addition operation is called a flop, and the total number of flops represents the computational burden.

For the considered system, the computational burden of DUKF is significantly lower than that of UKF. Specifically, it is necessary for UKF to select the sigma point set to derive the nonlinear system during the time update and measurement update. Consequently, UKF brings about a large but unnecessary computational burden because the time prediction equation of the considered system is linear. In contrast, DUKF uses KF in the linear part of the system, avoiding redundant calculations for measurement updates. Table 1 comprehensively compares UKF and DUKF in terms of the computational costs, especially for the time update part since they share the same measurement updates. It is deduced, through the analysis of the computational cost, that UKF requires a total of $\frac{n^{3}}{3}+17n^{2}+3n$ flops more than DUKF in each cycle, and they present comparative precision.

As shown in Figure 4, the calculation time of DUKF is in general concentrated below that of UKF. Table 2 shows that the variance of DUKF is 24.1% higher than that of UKF.

The 18% longer calculation time of DUKF than EKF is compensated by the much higher precision, and therefore we believe that the lengthened calculation time is worthwhile.

### 3.5. Improvement of the Stability of the DUKF

Compared with the traditional UKF algorithm, the DUKF algorithm adopted in this study is sensitive to the error of the initial system state estimation. Specifically, when the estimated initial system state is much different from the real counterpart, the DUKF algorithm is prone to divergence.

Due to the randomness of the covariance matrix, ill-posed problems are often encountered during the decomposition process, affecting the filtering performance and even causing divergence [25,26]. Countermeasures have been proposed in the study by Hu [20], who comprehensively analyzed the error characteristics and discussed the stability of the DUKF algorithm. According to his research, the observation covariance $P_{zz}$ is required to be a positive definite matrix to bound the estimation error of the DUKF algorithm. In this context, the noise variance $R_{k}$ needs to be a positive definite matrix as a precondition. However, $R_{k}$ may be a positive semi-definite matrix, which causes $P_{zz}$ to be a positive semi-definite matrix as well. To stabilize the algorithm, Hu added a small positive definite matrix $\Delta R_{k}$ to $R_{k}$:(6)$$R_{k}^{*}=R_{k}+\Delta R_{k}.\;$$

However, the introduction of $\Delta R_{k}$ subjects the algorithm to reduced filtering precision. In the study by Geng et al. [27], the SVD is introduced into the UKF algorithm to deliver a more complex iteration of the covariance matrix, which results in higher stability. Similarly, the SVD is introduced into the derived CKF algorithm to improve the stability in our previous research [28], which achieved good performance. In this study, the SVD is integrated with the previously introduced DUKF to form an SVD-DUKF method, followed by a simple proof and numerical experiments that verify its effectiveness.

Different from the Cholesky decomposition, the SVD can decompose positive semi-definite matrices. Therefore, $\Delta R_{k}$ is not needed anymore in the SVD-DUKF method. In other words, the stability of the algorithm can be guaranteed without increasing the measurement errors. This method not only optimizes the iterative process and reduces the amount of calculation, but also ensures the stability of the algorithm iteration, contributing to greatly improved system convergence.

### 3.6. The Method of the SVD-DUKF

Matrix *A*, assumed to be an m×n order matrix, can be decomposed into orthogonal matrices *U* and *V* according to the singular value decomposition theory:(7)$$A=U\Lambda V^{T},\Lambda =\left[\begin{array}{cc} S & 0\\ 0 & 0 \end{array}\right]\;$$
(8)$$S=diag\left(\sigma_{1},\sigma_{2},\ldots ,\sigma_{p}\right)\;$$
where $\sigma_{p}$ is the singular value of *A*, and the values are changed in a descending order.

When matrix *A* is a positive definite matrix, Formula (7) can be rewritten as
(9)$$A=USU^{T}.\;$$

In Formula (8), all of the $\sigma_{p}$ are greater than zero.

The error covariance matrix in the DUKF algorithm is subjected to the singular value decomposition, and the decomposition matrix is iterated by the properties of the matrix, which contributes to the improved SVD-DUKF algorithm.

Specific steps of the SVD-DUKF algorithm are detailed in Algorithm 1.
 **Algorithm 1.** The flow of the derivative SVD-DUKF algorithm.   **Input:**
${\hat{x}}_{k-1},{\hat{P}}_{k-1}$
   **Output:**
${\hat{x}}_{k},{\hat{P}}_{k}$
   1. Time update to the system based on the classical Kalman filter:${\hat{x}}_{k/k-1}=\varphi_{k/k-1}{\hat{x}}_{k-1}$${\hat{P}}_{k|k-1}=\varphi {\hat{P}}_{k-1}\varphi^{T}+Q_{k}$   2. SVD of state covariance matrix ${\hat{P}}_{k|k-1}$ can be obtained:${\hat{P}}_{k|k-1}=UD^{2}U^{T}$   3. Select a set of SIGMA points as follows:$\left\{\begin{array}{c} X_{i1|k}=X_{i|k}\\ X_{iu|k}=X_{i|k}\pm a\left(\left(\sqrt{n+\lambda }UD\right)\right) \end{array}\right.$   4. Update the observations based on the set of sigma points as follows:$Z_{iu|k+1}=H\left(X_{iu|k+1}\right)$$Z_{i|k+1}=\sum_{u=0}^{2n}\omega_{u}Z_{iu|k+1},\left(u=1,2,3,\ldots \right)$   5. Calculate the covariance of the observation:$P_{iziz|k+1}=\sum_{u=0}^{2n}\omega_{u}\left(Z_{iu|k}-Z_{i|k+1}\right){\left(Z_{iu|k}-Z_{i|k+1}\right)}^{T}$   6. Calculate the cross-covariance variance of observation and state:$P_{ixiz|k+1}=\sum_{u=0}^{2n}\omega_{u}\left(X_{iu|k}-X_{i|k+1}\right){\left(Z_{iu|k}-Z_{i|k+1}\right)}^{T}$   where $\left\{\begin{array}{c} \omega_{u}=1-\frac{1}{a^{2}},u=0\\ \omega_{u}=\frac{1}{2na^{2}},u=1,2,3,\ldots ,n \end{array}\right.$.   7. Obtain the Kalman gain:$K=P_{ixiz|k+1}P_{iziz|k+1}^{-1}$   8. Update the system status and covariance of the system:$\begin{array}{c} X_{i|k+1}=X_{i|k}+K\left({\hat{Z}}_{i}-Z_{i|k+1}\right)\\ P_{i|k+1}=P_{i|k}-KP_{ikik|k+1}K^{T} \end{array}$   9. **end.**

To demonstrate the performance of the SVD-DUKF algorithm, a simulation is implemented in a simple model, which demonstrates that the number of convergence of the SVD-DUKF algorithm is 56.7% higher than that of the DUKF algorithm (Table 3).

## 4. The Method of the Static Game Theory-Based SVD-DUKF Algorithm for Co-Localization

If we set robot *i* to be the observer and robot *j* to be observed in our one-way observation model, then robot *i* will provide the observation information, while robot *j* will select the strategy of localization, which depends on the information provided by robot *i*.

The observation information obtained by robot *i* concerning the location of robot *j* can be expressed as follows:(10)$$X_{j}^{\left(i\right)}=H_{ij}X_{i}=X_{i}+v_{k}\;$$
(11)$$\left[\begin{array}{c} x_{j}^{\left(i\right)}\\ y_{j}^{\left(i\right)} \end{array}\right]=H_{ij}\left[\begin{array}{c} x_{i}\\ y_{i} \end{array}\right]=\left[\begin{array}{c} x_{i}\\ y_{i} \end{array}\right]+v_{k},\;$$
where the superscript (i) means that the information is provided by robot *i*. Other parameters are listed in Figure 1. The SVD of the state covariance matrix $P_{i}$ can be obtained:(12)$$P_{i}=U_{i}D_{i}^{2}U_{i}^{T}.\;$$

Based on the result of the SVD and the previous independent localization, the sigma point set can be obtained:(13)$$\left\{\begin{array}{c} \xi_{0,i}=X_{i}\\ \xi_{u,i}=X_{i}+a\left(\sqrt{n+\lambda }U_{i}D_{i}\right),u=1,2,\ldots ,n\\ \xi_{u,i}=X_{i}-a\left(\sqrt{n+\lambda }U_{i}D_{i}\right),u=n+1,n+2,\ldots ,2n \end{array}\right.\;$$

After that, the sigma point set is transformed based on the measurement equation:(14)$$z_{ij}=H_{ij}\left(\xi_{u,i}\right),u=1,2,\ldots ,2n.\;$$

The predicted measurement and its covariance are then calculated:(15)$$z_{ij}=\sum_{u=0}^{2n}\omega_{u}\xi_{u,i},\;$$
(16)$$P_{z,ij}=\sum_{u=0}^{2n}\omega_{u}\left(\xi_{u,i}-z_{ij}\right){\left(\xi_{u,i}-z_{ij}\right)}^{T}+R_{ij}.\;$$

Subsequently, the cross-covariance between the state and the measurement can be calculated:(17)$$P_{xz,ij}=\sum_{u=0}^{2n}\omega_{u}\left(\xi_{u,i}-x_{ij}\right){\left(\xi_{u,i}-z_{ij}\right)}^{T},\;$$
where
$$\left\{\begin{array}{c} \omega_{0}=1-\frac{1}{a^{2}}\\ \omega_{u}=\frac{1}{2na^{2}},u=1,2,\ldots ,2n \end{array}\right.$$
*a* is an adjustment parameters.

The gain matrix *K* can be determined by the covariance matrix $P_{z,ij}$ and the cross covariance matrix $P_{xz,ij}$:(18)$$K_{ij}=P_{xz,ij}P_{z,ij}^{-1}.\;$$

Finally, based on the gain matrix $K_{ij}$ and the state covariance matrix $P_{i}$, the covariance matrix $P_{ij}$ of co-localization by robot *i* to robot *j* can be obtained:(19)$$P_{ij}=P_{i}+K_{ij}P_{z,ij}K_{ij}^{-1}.\;$$

The difference between $X_{j}^{\left(ij\right)}$ and $X_{j}$ is taken as the judging criteria. The eigenvalues of $\sqrt{P_{ij}}$, namely $a_{ij}$ and $b_{ij}$, are the semi-major and semi-short axes of the pose uncertainty ellipse, respectively. Similarly, by calculating the eigenvalues of $\sqrt{P_{ij}}$, namely $a_{ij}$ and $b_{ij}$, we have
(20)$$\frac{f\left(P_{ij},P_{j}\right)}{∥X_{j}^{\left(i\right)}-X_{j}∥}=\frac{\min\left\{3\times \sqrt{a_{ij}^{2}+b_{ij}^{2}},3\times \sqrt{a_{j}^{2}+b_{j}^{2}}\right\}}{\sqrt{{\left(x_{j}^{\left(i\right)}-x_{j}\right)}^{2}+{\left(y_{j}^{\left(i\right)}-y_{j}\right)}^{2}}}.\;$$

Based on Formula (20), the following benefit function is obtained:(21)$$f=\frac{\min\left\{3\times \sqrt{a_{ij}^{2}+b_{ij}^{2}},3\times \sqrt{a_{j}^{2}+b_{j}^{2}}\right\}}{\sqrt{{\left(x_{j}^{\left(i\right)}-x_{j}\right)}^{2}+{\left(y_{j}^{\left(i\right)}-y_{j}\right)}^{2}}}.\;$$

The game benefits of *i* and *j* are shown in Table 4. Moreover, in the case of the one-way observation, robot i performs autonomous localization and the benefit of the game is 1. If f>1, $u_{j}\left[s_{i},s_{j}=1\right]=f>u_{j}\left[s_{i},s_{j}=0\right]=1$, the game can reach the unique Nash equilibrium (independent localization; co-localization). If f<1, $u_{j}\left[s_{i},s_{j}=1\right]=f<u_{j}\left[s_{i},s_{j}=0\right]=1$, the game can reach the unique Nash equilibrium (independent localization; independent localization). If f=1, the benefit function of the game is always equal to 1. Co-localization tends to increase the computational burden, and independent localization is thus generally selected by the robots.

After identifying the selection strategy of the game with two robots in the case of one-way relative observation, the proposed SVD-DUKF algorithm is adopted to obtain the location information of each robot. Specific steps of the SVD-DUKF algorithm are shown in Algorithm 1.

Finally, the calculation results are assigned to the corresponding robots participating in the co-localization. Specific steps of the game theory-based SVD-DUKF co-localization algorithm (with localization of robot *j* as an example) are shown in Algorithm 2. The flow chart of the algorithm is shown in Figure 5.
 **Algorithm 2.** The flow of the game theory-based SVD-DUKF algorithm.   **Input:**
$x_{i|k}$, $P_{i|k}$, $x_{j|k}$, $P_{j|k}$   **Output:**
$x_{i|k+1}^{\left(j\right)}$, $P_{i|k+1}$, $x_{j|k+1}^{\left(i\right)}$, $P_{j|k+1}$   1. Calculate the $x_{i|k+1}$, $P_{i|k+1}$, $x_{j|k+1}$, $P_{j|k+1}$ by Algorithm 1.   2. Calculate the $a_{ji}$, $b_{ji}$, $a_{j}$, $b_{j}$ by the SVD.   3. Substitute the above results into the benefit function (Formula (21)).   4. Based on the result of Formula (21), choose the game strategy.   5. Based on the strategy, **return** the $x_{i|k+1}^{\left(j\right)}$, $P_{i|k+1}$, $x_{j|k+1}^{\left(i\right)}$, $P_{j|k+1}$.   6. **end.**


## 5. Simulations

In order to demonstrate the performance of the proposed algorithm, a MATLAB simulation is implemented based on the open-source program that is provided by OpenSLAM [29].

As shown in Figure 6, the connection between the asterisk and the triangle indicates that the UWB sensor (asterisk) is measuring the distance of the robot (triangle). All robots can observe the relative angle and distance of the objects in a range of 180 degrees in front of them, with a maximum observation range of 15 m, and the paths they move are represented by green, purple, and blue lines. The simulation space is a 200 m × 200 m square. The dynamic model of the robot and the error variance of the sensor are determined by the system model proposed in Section 3.3.

The SVD-DUKF algorithm is adopted when each robot is localized individually, and the optimized game theory-based algorithm is used when a relative observation occurs.

To demonstrate the performance of the proposed algorithm, five algorithms are evaluated by the simulations, including the original EKF, the static game theory-based EKF, the static game theory-based UKF, and the static game theory-based SVD-DUKF. Moreover, another nonlinear filtering algorithm (square root cubature Kalman filter, SR-CKF) proposed by Liu et al. [30], which is also of higher precision and convergence, is assessed as the control group. The simulation results are shown in Figure 7, Figure 8, Figure 9 and Figure 10.

Localization errors of the other four algorithms are all less than that of the original EKF (Figure 7). Moreover, the game theory-based UKF has fewer localization errors than the original EKF and the game theory-based EKF (Figure 8). In addition, the SVD-DUKF algorithm further reduces the localization error (Figure 9). As shown in Figure 10, the SR-CKF as a control group also presents higher performance than the game theory-based EKF, and it performs similarly well with the algorithm proposed in this research.

However, due to the introduction of the derivative method, the game theory-based SVD-DUKF proposed in this paper is subjected to a smaller computational burden than SR-CKF, which is proven in the average calculation time of 50 times of simulations (Table 5). Figure 11 and Figure 12 show the average localization errors and variances of all the five algorithms. Consistent with the previous analyses, the game theory-based SVD-DUKF presents better performance than other original algorithms and UKF series algorithms. Its average error is almost the same as that of SR-CKF, but its variance is much smaller than that of SR-CKF. Therefore, it is safe to conclude that the algorithm proposed in this study has the best performance among the five algorithms.

It is confirmed by the simulation results that SR-CKF and SVD-DUKF have better performances than the others. Moreover, as has been stated in Section 2, UKF has higher precision than the game theory-based EKF, which explains the poor performance of EKF in the simulation. Different from the original UKF, SVD-DUKF adopts the linear update in the time update, thus removing the linearization error. Therefore, SVD-DUKF has higher precision than the original UKF in the simulations. Although SR-CKF also presents very high precision in the simulation, it needs to calculate the volume point set during the time update phase, thereby resulting in a much heavier calculation burden than SVD-DUKF. Consequently, SVD-DUKF is believed to outperform SR-CKF, although they have almost the same precision and variance.

## 6. Conclusions

The main purpose of this study is to improve a game theory-based co-localization algorithm that is filtered by EKF. After introducing the advantages of UKF in terms of precision, the filtering part of the original game theory-based co-localization algorithm is modified. Moreover, this study manages to relieve the computational burden and to enhance the algorithm stability through combining DUKF with SVD-UKF, and the combination is utilized in the game theory-based co-localization algorithm. Simulation results show that the proposed algorithm can increase the co-localization precision, without increasing the computational burden and deteriorating the stability. Therefore, it is safe to conclude that the proposed algorithm performs well in the scenarios explored in this study, and it can effectively solve the information conflicts caused by multi-sensors in the wireless sensor network.

We aim to perform the following in the future:test the performance of the proposed algorithm on the physical platform, rather than only through numerical simulation;since the CKF series algorithms have better stability and precision theoretically, apply similar improvements to the CKF series algorithms in subsequent research;apply the proposed algorithm to SLAM;testing the performance of the optimized algorithm on a complex dynamic model.

## Figures and Tables

**Figure 1 sensors-20-05579-f001:**
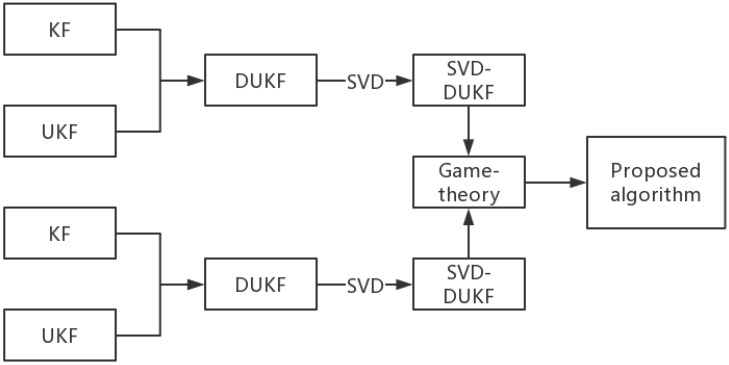
Schematic diagram of the relative observation.

**Figure 2 sensors-20-05579-f002:**
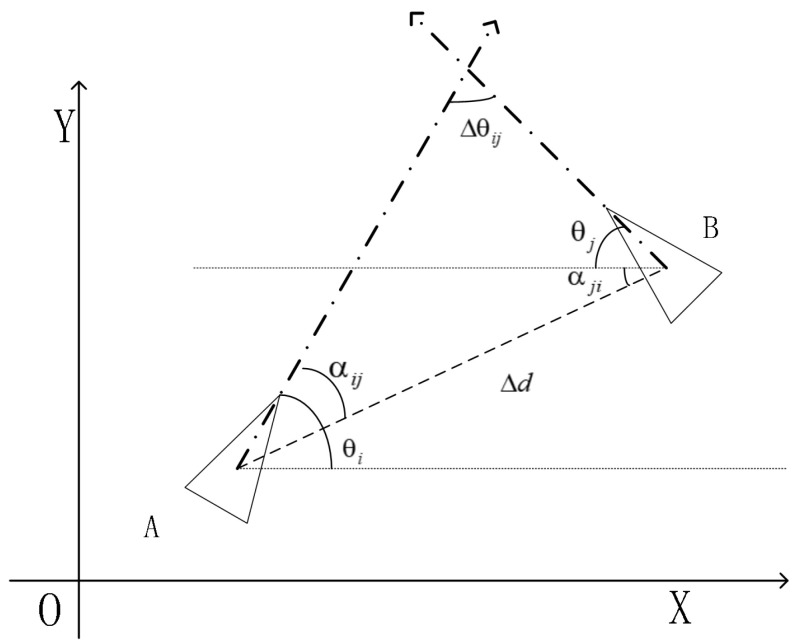
Schematic diagram illustrating the relative observation between two robots.

**Figure 3 sensors-20-05579-f003:**
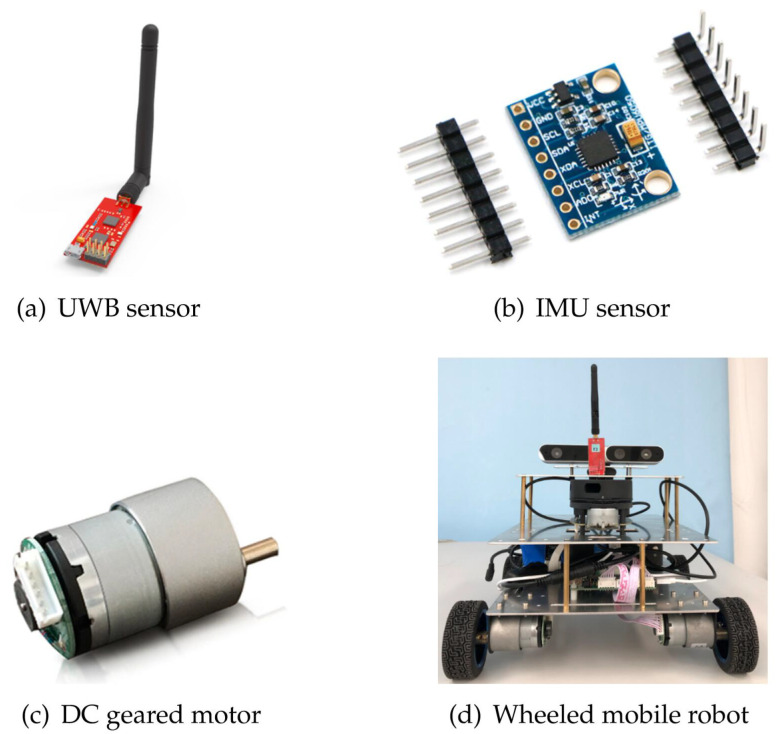
The sensors and the mobile robot involved in this article. (**a**) The Ultra-Wide Band (UWB) sensors(YCHIOT, Wenzhou, Zhejiang, China), which are commercial products provided by the company of YCHIOT, with the module of Mini3s; (**b**) the spatial motion sensor chip MPU6050 as the IMU sensor(Digi-Key Electronics, Thief River Falls, MN, US); (**c**) the odometer84 constructed by the DC gear motors MG513 with an encoder(Fenghua Transmission, Kunshan, Jiangsu, China); (**d**) the wheeled mobile robot platform(Ruiqu Technology, Foshan, Guangdong, China) that realizes the precise localization by carrying the above sensors.

**Figure 4 sensors-20-05579-f004:**
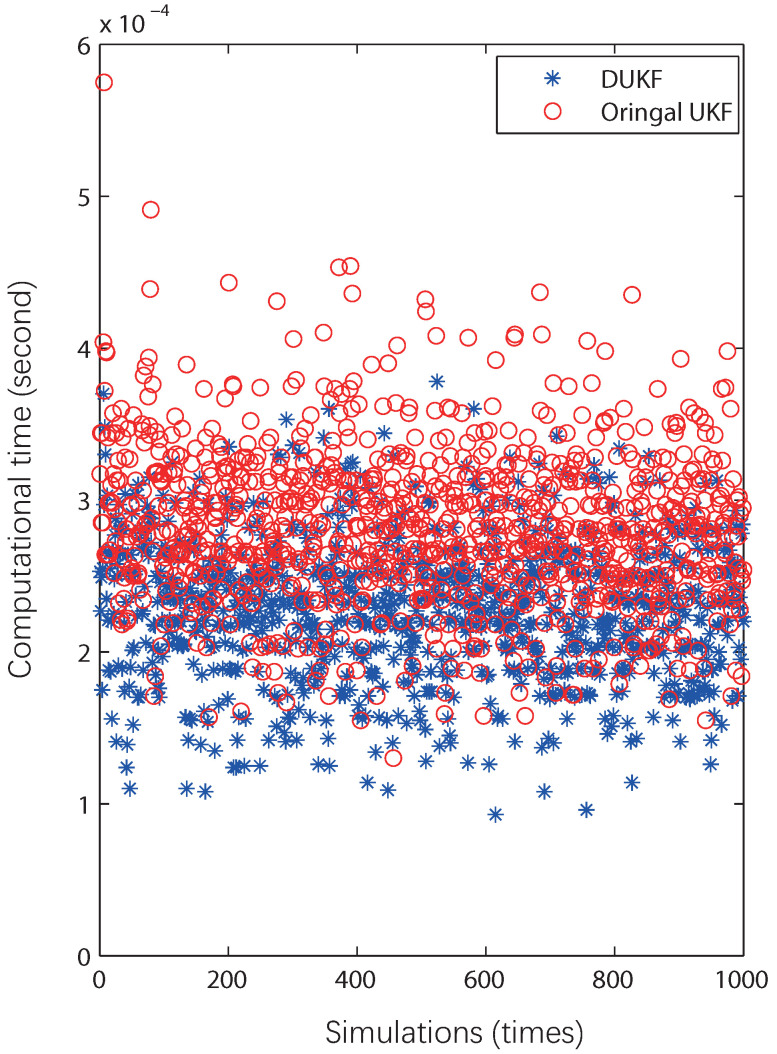
The computational time of UKF and DUKF during the 1000 Monte Carlo simulations of a simple model, where blue asterisks represent the calculation time of DUKF, while the red circles stand for that of UKF.

**Figure 5 sensors-20-05579-f005:**
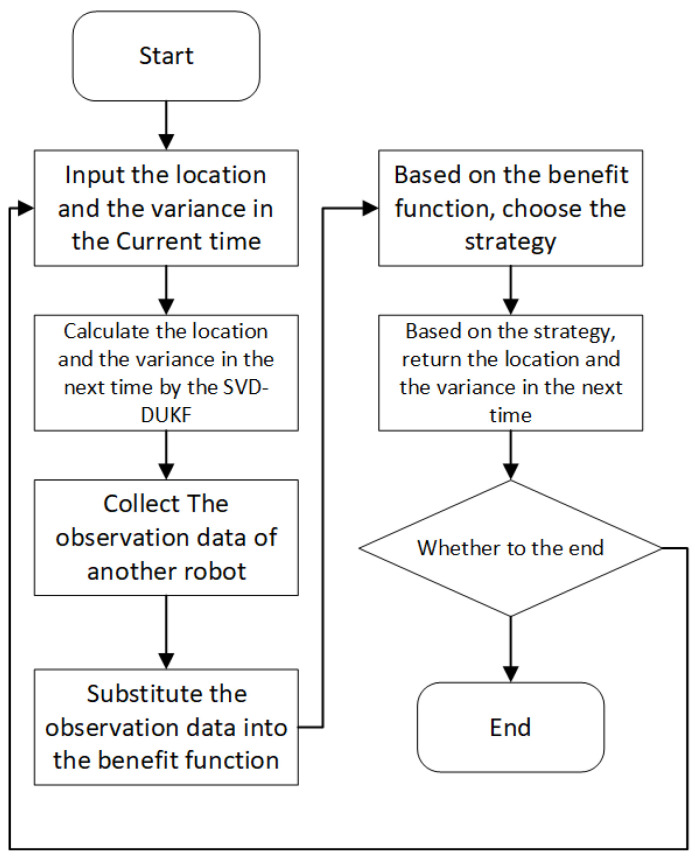
The flow chart of the static game theory algorithm based on SVD-DUKF.

**Figure 6 sensors-20-05579-f006:**
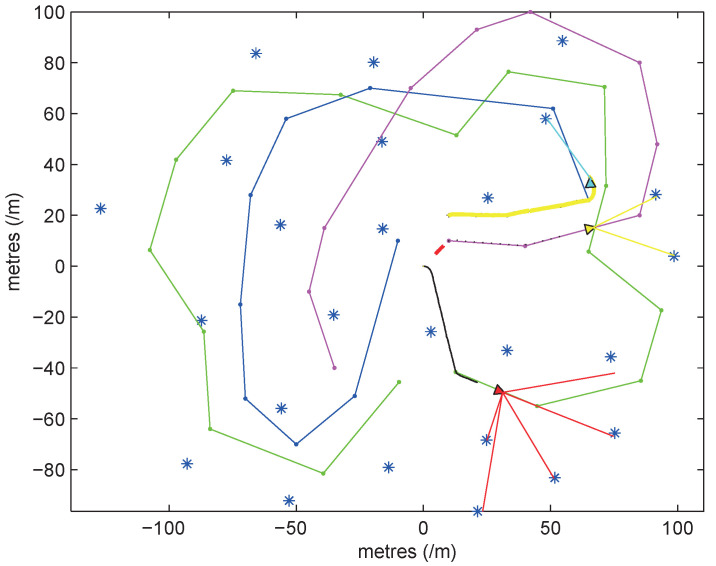
Simulation space.

**Figure 7 sensors-20-05579-f007:**
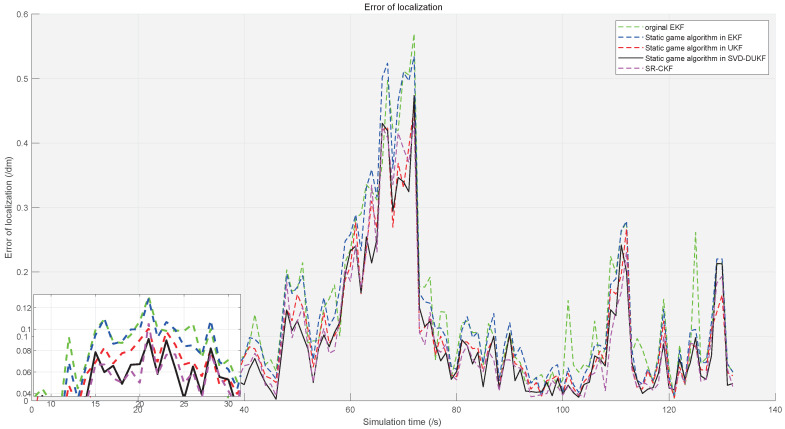
Localization errors of the five algorithms, with the small box in the lower left corner as a partially enlarged view. The control group and the original algorithm are represented by dashed lines in various colors, and the proposed algorithm is represented by the black solid line.

**Figure 8 sensors-20-05579-f008:**
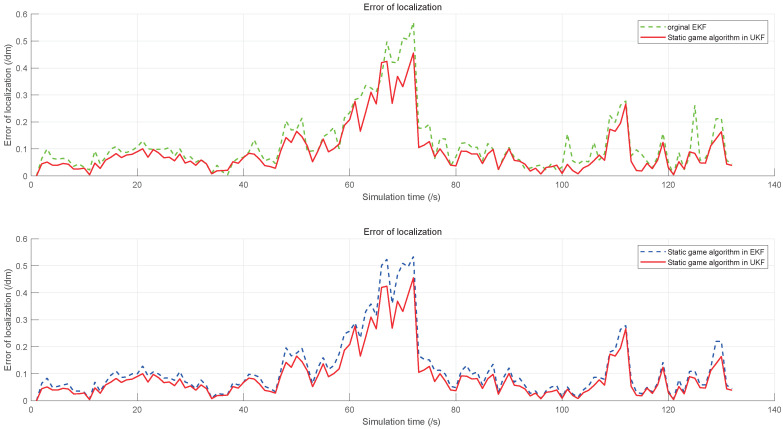
The error of localization on the game theory UKF, the original extended Kalman filtering (EKF), and the game theory EKF.

**Figure 9 sensors-20-05579-f009:**
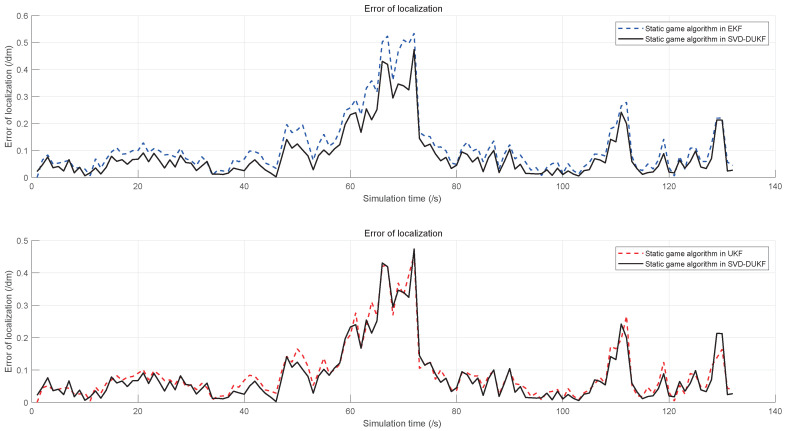
The error of localization on the game theory EKF, the game theory UKF, and the game theory SVD-DUKF.

**Figure 10 sensors-20-05579-f010:**
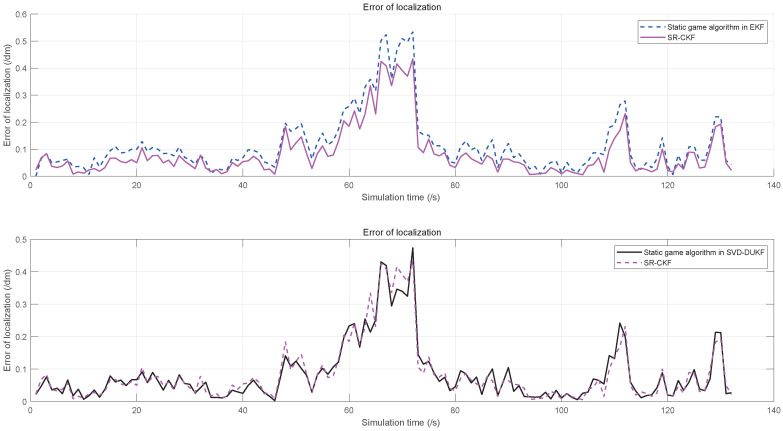
The error of localization on the game theory EKF, square root cubature Kalman filter (SR-CKF), and the game theory SVD-DUKF.

**Figure 11 sensors-20-05579-f011:**
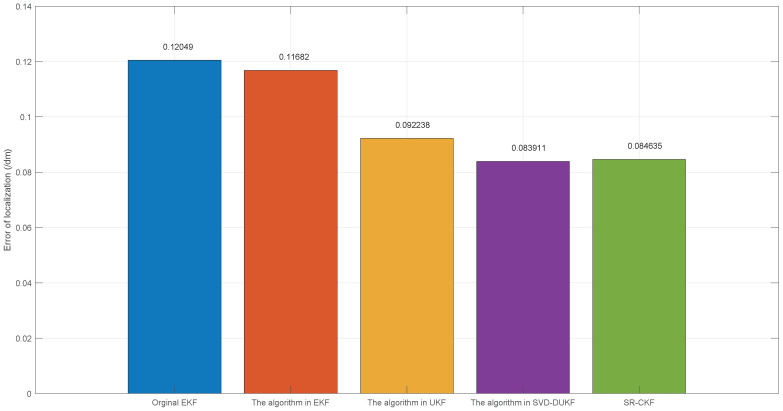
The mean of the localization error.

**Figure 12 sensors-20-05579-f012:**
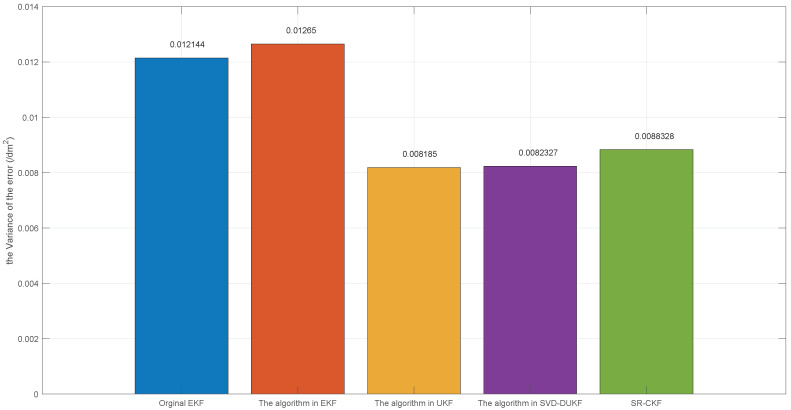
The variance of the localization error.

**Table 1 sensors-20-05579-t001:** The difference in Computation Cost of unscented Kalman filter (UKF) and derivative UKF (DUKF).

Step	UKF	DUKF	Flops Difference
Calculate the Sigma points (Time update)	Cholesky decomposition to the $P_{i|k}$	none	$\frac{n^{3}}{3}+n^{2}$
	$\left\{\begin{array}{c} X_{i1|k}=X_{i|k}\\ X_{iu|k}=X_{i|k}\pm a\left(\sqrt{nP_{i|k}}\right) \end{array}\right.$		$3n^{2}$
Predict the time update of Sigma points	${\hat{X}}_{iu|k+1}=\varphi \left(x_{iu|k}\right)$	none	$n^{2}$
Estimate the time update	${\hat{X}}_{k+1|k}=\sum_{2n+1}^{u=1}\omega_{u}{\hat{X}}_{iu|k+1}$	${\hat{X}}_{k+1|k}=\varphi {\hat{X}}_{k}$	$4n^{2}+n$
	$P_{k+1|k}=\sum_{2n+1}^{u}\omega_{u}\left({\hat{X}}_{k+1|k}-X_{iu|k+1}\right){\left({\hat{X}}_{k+1|k}-X_{iu|k+1}\right)}^{T}+Q_{k}$	$P_{k+1|k}=\varphi P\varphi^{T}+Q_{k}$	$8n^{2}+2n$

**Table 2 sensors-20-05579-t002:** Variance of the UKF and the DUKF.

Algorithm	Variance
UKF	0.2282
DUKF	0.2832

**Table 3 sensors-20-05579-t003:** The success convergence rate of the DUKF and the singular value decomposition (SVD)-DUKF in 1000 simulations.

Algorithm	Convergence Rate
DKF	43.7%
SVD-DUKF	100%

**Table 4 sensors-20-05579-t004:** Table of game benefits.

Strategy	$s_{j}=1$	$s_{j}=0$
$Ds_{i}=1$	$f_{i},f_{j}$	$f_{i}$,1
$s_{i}=0$	1,$f_{j}$	1,1

**Table 5 sensors-20-05579-t005:** The average calculation time of SR-CKF and the static game theory-based SVD-DUKF in 50 simulations.

Algorithm	Average Calculation Time(s)
static game theory SVD-DUKF	21.6658
SR-CKF	24.8356

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
