# Peer review of "An Improved Game Theory-Based Cooperative Localization Algorithm for Eliminating the Conflicting Information of Multi-Sensors"

_sensors, 2020, doi:10.3390/s20195579_

Round 1

Reviewer 1 Report

The paper presents an algorithm combining UKF and game-theory for multi-robot systems' localization. The SVD is proposed to replace Cholesky decomposition to improve the stability. UKF is proposed to replace EKF. The paper is not well written, which is the biggest problem of the paper. The English errors are too much so that it is annoying when reading the paper. The organization of the paper is also problematic. Figure 4 shows the real robot. However, the performance verification is only done in simulation. Why do you need to introduce the hardware of the robot? Also, the literature is not well reviewed. The cooperative localization problem has been well solved in the literature given the conditions described in the simulation of the multi-robot system. Another doubtful thing is that the system should be a linear system in the simulation based on Eq. (18), the system state space equation. Why do you need UKF for a linear system?

I suggest the paper should be rewritten and improved by a native English speaker. Otherwise, the paper is too weak for publication.

Reviewer 2 Report

The content of the article is consistent with the scientific area of the Journal - Sensors. The subject raised by the authors is current and so far rarely noticed by other authors publishing in this area.
The issue described may in the future contribute to improving the efficiency of the automation and of the eliminating conflicting information of multi-sensor.
In this study, the SVD method is introduced into the previously introduced DUKF to form a SVD-DUKF method and conduct simple proof and numerical experiments to verify its effectiveness. However, the time predict equation of the system considered in this article is a linear equation, and choosing this method obviously increases the computational burden. The paper has an original, scientific character, related to an Improved Game-Theory based Cooperative of Multi-sensor.
In order to demonstrate the performance of the algorithm proposed in this paper, a MATLAB simulation is performed based on the SLAM open source simulation program, which is provided by OpenSLAM.
For a better clarification, please edit your paper as follows: 1. Extend the introduction to current results in the world and in Europe, - links to the dissemination of the results of European authors, articles registered in SCOPUS / WoS, such as: Reliability determination and diagnostics of a mechatronic system. Experimental investigations of a highly maneuverable mobile omniwheel robot, Integration of Inertial Sensor Data into Control of the Mobile Platform, Navigation control and stability investigation of a mobile robot based on a hexacopter equipped with an integrated manipulator. Figure 6 should be contrasting and readable. Conclusions and future work should be extended to contain practical applications based on research described in this paper - expand references. Highlight the course of dependencies / relations in figures No: 7 - 10. The paper should be read by a native english speaker.
I recommend publishing the post after the proposed modifications.

Round 2

Reviewer 1 Report

I can see the improvement of the English of the paper. But there are still some grammar errors, for example, line 11, 64, 68, etc. Also, in the Abstract, please explain how the filtering process is optimized. Otherwise, it is confusing after reading the Abstract. 
